# Meta-Learning with Personalized Learning Rates for Rapid Task Mastery

## Abstract

Traditional meta-learning approaches primarily focus on the generalization ability of models across unfamiliar tasks. These methods typically involve fine-tuning the model in the outer loop to perform well on new tasks. While this is valuable for enabling models to adapt to various tasks, it may overlook the details of rapid adaptation within tasks. During rapid adaptation, the same task may exhibit entirely different data distributions, features, and patterns in different training phases, making it exceptionally challenging to determine an appropriate learning rate. Consequently, conventional meta-learning methods often employ fixed learning rates or simple learning rate strategies, overlooking the dynamic nature within tasks. In this paper, we propose an Meta-Learning with Personalized Learning Rates (MLPLR) approach. Specifically, we adaptively generate negatively correlated learning rates by evaluating the information loss between predicted values and ground truth. When the information loss is low, indicating the model's strong performance on the current task, we can increase the learning rate to expedite the learning process. This aids in faster convergence and adapting to specific patterns and features within tasks. Conversely, when the information loss is high, indicating poor model performance on the current task, we reduce the learning rate to ensure more stable and gradual parameter updates, thereby mitigating overfitting. Extensive experiments and analyses demonstrate that our approach enhances the performance of various meta-learning models in the contexts of few-shot classification, few-shot fine-grained classification, and cross-domain few-shot classification.

## 1 Introduction

In an era characterized by the rapid advancement of artificial intelligence technology, deep learning algorithms have found wide-ranging applications across various domains (Liu et al., 2016; He et al., 2016; Nam & Han, 2016; Chen et al., 2017). However, these algorithms typically demand substantial volumes of data and formidable computational resources to perform effectively. Yet, real-world scenarios often impose constraints on the availability of data and computational capacity. To address this challenge, meta-learning (Schmidhuber, 1987; Thrun & Pratt, 2012) emerges as a highly anticipated solution. At the heart of meta-learning lies the fundamental concept of extracting general knowledge and strategies from prior learning experiences, empowering machines to swiftly adapt to new tasks. This implies that even when confronted with limited data samples, machines can achieve favorable outcomes rapidly.

An important branch of meta-learning is metric-based meta-learning (Vinyals et al., 2016; Snell et al., 2017; Sung et al., 2018), where machine learning focuses on meta-learning by measuring the similarity between tasks. Another significant branch of meta-learning is optimization-based meta-learning(Finn et al., 2017; Nichol et al., 2018; Raghu et al., 2019), in which machine learning adapts to new tasks through an optimization process. This typically involves bi-level optimization (Ravi & Larochelle, 2016): outer optimization (Baik et al., 2020b; Grant et al., 2018; Finn et al., 2018; Jamal & Qi, 2019; Vuorio et al., 2019; Yao et al., 2019) and inner optimization (Behl et al., 2019; Lee & Choi, 2018; Li et al., 2017; Rusu et al., 2018). Outer optimization is used to learn how to initialize model parameters for rapid adaptation to new tasks. Inner optimization, on the other hand, fine-tunes model parameters on new tasks to achieve improved performance. These bi-level optimization collaborate to accomplish the objectives of meta-learning. Unlike metric-based meta-

learning, optimization-based meta-learning emphasizes how to rapidly adapt to new tasks through optimization algorithms, rather than considering the similarity between task instances.

Optimization-based meta-learning approaches (Zucchet et al., 2022; Zhao et al., 2022; Chen et al., 2022; Li et al., 2023; Sendera et al., 2023) do not rely on specific model architectures but rather concentrate on learning general optimization strategies. This flexibility enables them to be applicable to a variety of deep learning models, free from the constraints of particular model structures. Consequently, optimization-based meta-learning methods are typically more versatile.

A prominent representative of optimization-based meta-learning is MAML (Model-Agnostic Meta-Learning) (Finn et al., 2017). MAML introduces a universal optimization strategy that enables models to perform well on new tasks with only a few gradient updates. Although MAML is a powerful meta-learning method, it has its share of shortcomings and limitations (Rajendran et al., 2020; Yin et al., 2019). To address challenges arising from diverse tasks and data distributions, recent research has started to consider the introduction of scheduling strategies in task sampling, effectively constructing classification tasks through adaptive sampling processes (Liu et al., 2020a; Yao et al., 2021b;d). In recent studies, the application of task augmentation methods (Rajendran et al., 2020; Yao et al., 2021c; Wu et al., 2022; Liu et al., 2020b; Yao et al., 2021a) has significantly improved model performance in addressing issues caused by insufficient tasks. However, these strategies represent just the tip of the iceberg in the field of meta-learning. Recent research suggests that methods involving adaptive learning rates can also enhance model performance and robustness.

In traditional gradient descent algorithms, the learning rate is typically set as a static value, which must be manually adjusted before training to perform well across various tasks and datasets. However, this "one-size-fits-all" approach often struggles to strike a balance between training speed and convergence quality. Adaptive learning rates (Zou et al., 2021; Yu et al., 2021; Kim et al., 2022; Behl et al., 2019), as a meta-learning strategy, aim to enable models to dynamically adjust the learning rate during different tasks and learning stages. This capability allows models to better adapt to task complexity and data variability, ultimately leading to improved generalization.

However, in existing adaptive learning rate methods, many approaches require the introduction of additional computational resources to achieve adaptivity. This may involve extra model parameters, computationally expensive iterative processes, or complex hyperparameter tuning. Therefore, we propose an adaptive learning rate method that does not require the introduction of additional computational resources. Specifically, we dynamically generate negatively correlated learning rates by evaluating the information loss between the model's predictions and the ground truth. The generation of these negatively correlated learning rates adapts to the current task based on its performance. When the information loss is low, indicating that the model is performing well on the current task, we can increase the learning rate to accelerate parameter updates and the learning process. This helps the model converge faster and adapt better to specific patterns and features within the task, thus improving training efficiency and performance. Conversely, when the information loss is high, indicating poor model performance on the current task, we reduce the learning rate to slow down parameter changes to avoid overfitting and unstable parameter updates. This makes parameter updates more stable and gradual, aiding the model in better adaptation to challenging tasks while reducing the risk of overfitting.

This method for generating adaptive learning rates allows the model to employ different learning rate strategies at various stages, aligning itself more effectively with the requirements of the task. Such a strategy enhances the model's performance across diverse tasks and data conditions while reducing the need for hyperparameter tuning, making the training process more intelligent and efficient.

In summary, the main contributions of our work are as follows.

1. We propose a Meta-Learning with Personalized Learning Rates, designed to accommodate dynamic variations in within-task rapid adaptation.

2. Our method stands out for its independence from additional computational resources, making it effortlessly applicable across various computing environments without the need for increased hardware or software costs.

3. Extensive experimentation has demonstrated the wide applicability of our method, irrespective of model constraints. It seamlessly integrates with various meta-learning models, thereby enhancing their generalization performance.

The rest of this paper is organized as follows. In Section 2 we review previous work related to our approach. Section 3 elaborates on the details of our proposed meta-learning method for feature distribution alignment. Section 4 presents the experimental setup and results under different scenarios. Finally, conclusions are given in Section 5.

## 2 RELATED WORK

Meta-learning, as a captivating field, is undergoing a vibrant development, aiming to address the challenges of learning with limited data. Currently, meta-learning methods can be broadly categorized into two main classes: metric-based methods and optimization-based methods.

Metric-based meta-learning methods perform classification by measuring the similarity between example features and category centroids, with the objective of acquiring a shared distance metric for cross-task applicability. One of the pioneering metric-based meta-learning approaches is Matching Networks (Vinyals et al., 2016), which achieves rapid adaptation and classification through sample encoding, learning mapping functions, and the incorporation of attention mechanisms. Prototypical Networks (Snell et al., 2017), on the other hand, map input samples into a shared embedding space and utilize prototype vectors for classification predictions. Conversely, Relation Networks (Sung et al., 2018) express sample relationships through relation modules, avoiding direct involvement with embedding spaces.

Optimization-based meta-learning methods, on the other hand, aim to enable rapid and efficient adaptation to new tasks by iteratively updating global model parameters through multiple gradient descent steps. Model-Agnostic Meta-Learning (MAML) (Finn et al., 2017) stands out as a representative in this category due to its concise and versatile design. To mitigate the computational complexity of MAML, researchers have introduced FOMAML, which substitutes tedious second-order gradient calculations with a first-order approximation. Reptile (Nichol et al., 2018) takes a further simplifying approach by utilizing the difference in approximate derivatives between parameter estimates and initial values. Additionally, researchers have observed that MAML's effectiveness primarily stems from feature reuse, leading to the development of ANIL (Raghu et al., 2019), a method that requires minimal internal loop updates and performs on par with MAML in terms of performance.

Optimization-based methods focus on adjusting the optimization algorithms themselves to make them widely applicable across various domains, offering significant potential for improvement. Hence, this paper primarily investigates gradient-based meta-learning methods. While MAML and its variants have achieved notable successes, there are still some potential limitations. MetaOpt-Net (Lee et al., 2019) replaces the linear predictor with a support vector machine, and researchers like Bertinetto et al. (Bertinetto et al., 2018) have constructed differentiable closed-form solvers to further enhance performance. Recent research has started exploring the possibility of introducing scheduling strategies in task sampling to more effectively construct classification tasks (Liu et al., 2020a; Yao et al., 2021b). Moreover, the application of task augmentation methods has significantly improved model performance in addressing issues caused by insufficient tasks. These methods include injecting identical random noise into task labels (Rajendran et al., 2020), using task interpolation techniques (Yao et al., 2021c), adversarial task oversampling (Wu et al., 2022), image rotation strategies (Liu et al., 2020b), and mixing different task instances (Yao et al., 2021a), among others.

Although the above-mentioned methods have excelled in achieving impressive results, they seem to overlook a crucial aspect, which is the variation in the model's learning rate across different tasks and learning stages. To address this issue, different methods have employed various strategies to adjust learning rates and decay rates. MAML++ (Antoniou et al., 2018) uses step functions or cosine functions for learning rate annealing to adaptively adjust learning rates during different learning stages. Meta-LSTM (Ravi & Larochelle, 2016) takes a more complex approach by using an LSTM (Hochreiter, 1998) network as the outer network to learn optimization parameters for the inner network, including learning rates and decay rates. While this method can achieve dynamic learning rates and decay rates, its feasibility in practical applications is lower due to the complexity of LSTM training and its slower convergence. In contrast, Meta-SGD (Li et al., 2017) simplifies the learning rate adjustment process by using the model's own output results for learning rate updates, improving practicality.

Another approach is ALFA (Baik et al., 2020a), which designs a novel update rule to dynamically generate learning rates and weight decay terms for each update step and task based on the gradients and weights of the base learner. However, some of these dynamic learning rate methods may be too simplistic to adapt well to complex tasks and data distributions, while others may require additional computational resources or complex model structures. In practical applications, selecting an appropriate learning rate adjustment method that suits specific tasks and resource constraints is crucial.

Therefore, we propose a meta-learning method for adaptive learning rates that does not require the introduction of additional computational resources. In the inner loop learning of meta-learning, we dynamically generate negatively correlated learning rates by evaluating the information loss between the model's predictions and the ground truth. What sets this method apart is that it does not alter the existing structure and time complexity of meta-learning models while achieving outstanding results in terms of performance. Furthermore, its simplicity allows for easy integration with other algorithms.

## 3 PROPOSED METHOD

### 3.1 PRELIMINARIES

Meta-learning seeks to discover a model denoted as $f_\theta$ capable of fast learning and adaptation across a diverse range of tasks. Specifically, in the context of few-shot classification tasks, these tasks are often framed as $N$-way $K$-shot classification. Here, $N$ refers to the number of classes within each task, while $K$ signifies the number of available samples for each class.

In this setup, we assume the existence of a task distribution, denoted as $p(\mathcal{T})$, from which we draw a mini-batch $\{\mathcal{T}_1, \mathcal{T}_2, \cdots, \mathcal{T}_B\}$ with a batch size of $B$. Staying consistent with the conventions of $N$-way $K$-shot classification experiments, for each task $\mathcal{T}_i$, we gather $N \times K$ samples to construct a support set $\mathcal{D}^s = \{X^s, Y^s\}$ and $N \times M$ samples to create a query set $\mathcal{D}^q = \{X^q, Y^q\}$. Importantly, it should be noted that there is no overlap between the $\mathcal{D}^s$ and the $\mathcal{D}^q$.

In the context of meta-learning, parameter updates involve a two-stage optimization process. It commences with the model learning from its initial parameters denoted as $\theta$. In the inner loop, the support set of each task plays a pivotal role by facilitating rapid adjustments of model parameters or the computation of tailored distance metrics that align with the specific task requirements. Simultaneously, the outer loop focuses on updating the parameters of the meta-model, thereby enhancing the model's capacity for quick adaptation and learning across a variety of tasks.

To formally describe optimization-based meta-learning, we execute a sequence of gradient descent steps in the inner loop, employing $j$ steps where $j = \{1, 2, \cdots, k\}$. Each step is updated according to the following procedure:

$$\theta_{i,j+1} = \theta_{i,j} - \beta \nabla_\theta \mathcal{L}_{\mathcal{T}_i}^{\mathcal{D}^s}(f_{\theta_{i,j}}) \tag{1}$$

Where $\beta$ is the inner loop learning rate. Following the acquisition of inner loop parameters $\theta_i' = \theta_{i,j+1}$ for the $B$ tasks, the outer loop employs the query set for forward propagation. During this process, it computes the average cross-entropy loss for these $B$ tasks. The calculation of the average cross-entropy loss can be described as follows:

$$\mathcal{L}^{\mathcal{D}^q} = \frac{1}{B} \sum_{i=1}^{B} \mathcal{L}_{\mathcal{T}_i}^{\mathcal{D}^q}(f_{\theta_i'}) \tag{2}$$

The final model's parameters undergo updating through the following procedure:

$$\theta \leftarrow \theta - \alpha \nabla_\theta \mathcal{L}^{\mathcal{D}^q} \tag{3}$$

Where $\alpha$ is the outer loop learning rate.

### 3.2 META-LEARNING WITH PERSONALIZED LEARNING RATES

In the context of bi-level optimization, meta-learning's outer optimization is used to learn how to initialize model parameters, while the inner optimization fine-tunes model parameters on new tasks. In previous methods, the training speed for each inner-loop task was consistent. However, due to variations between different tasks at different stages, this uniform training strategy is not ideal. Although some algorithms have started introducing adaptive learning rates to achieve some success, these methods often require the introduction of additional model parameters, costly computational iterations, or complex hyperparameter tuning, which may not be practical in real-world scenarios. Therefore, we propose a personalized learning rate approach to address the dynamic changes in rapid adaptation within tasks without the need for additional computational resources. The overall framework is illustrated in Figure 1.

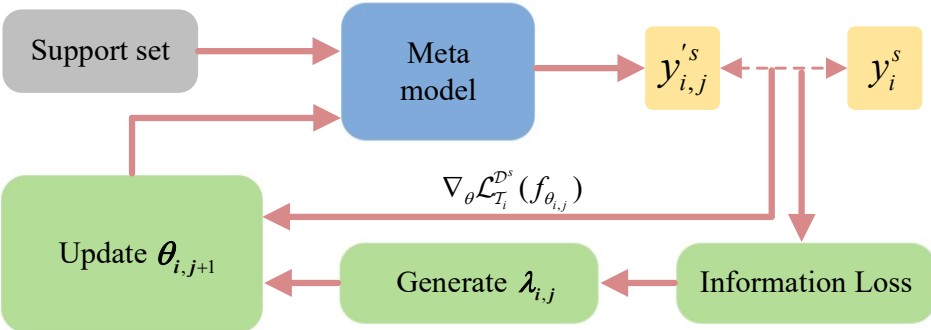

Figure 1: The overall framework of our proposed MLPLR. Firstly, through forward propagation, obtain the predictions (denoted as $y_{i,j}^{\prime s}$) for the task and employ the information loss between these predictions and the ground truth as the evaluation metric. Next, generate personalized learning rates based on the information loss and use them along with the loss gradients on the support set for the parameter updates within the inner loop.

Algorithm 1 summarizes the proposed personalized learning rates approach. First, the task's predictions, denoted as $y_{i,j}'$, are obtained through forward propagation. Then, the information loss between these task predictions $y_{i,j}'$ and the ground truth $y_i$ is employed as the evaluation metric. In this assessment, we choose the Kullback-Leibler (KL) divergence as the measure, where its magnitude reflects the performance of the current task. Information loss calculation is based on the following equation:

$$\mathcal{L}_{KD_{i,j}} = \mathcal{D}_{KL}(p(y_{i,j}')\|p(y_i)) \tag{4}$$

The magnitude of the KL divergence provides an intuitive reflection of the current task's performance. A larger KL divergence value implies relatively poorer model performance on the current task, while a smaller KL divergence suggests good performance on the current task. To personalize the learning rate further, negative correlated learning rates are generated based on the current task's KL divergence. This negative correlation ensures an inverse relationship between the learning rate and performance, meaning that the learning rate decreases when performance is poor and increases when performance is good, thus adapting more effectively to task variations. The variation in learning rates is depicted in Figure 2.

However, to prevent excessively large or small learning rates from adversely affecting model performance, it is necessary to impose constraints on the learning rate to ensure it stays within a reasonable range. These constraints involve setting an upper limit, denoted as $lr_{max}$ and a lower limit, denoted as $lr_{min}$ for the learning rate. The final computation for the adaptive learning rate $\lambda_{i,j}$ is as follows:

$$\lambda_{i,j} = \begin{cases} lr_{max} - \mathcal{L}_{KD_{i,j}}, & if \mathcal{L}_{KD_{i,j}} \le lr_{max} \\ lr_{min}, & if \mathcal{L}_{KD_{i,j}} > lr_{max} \end{cases} \tag{5}$$

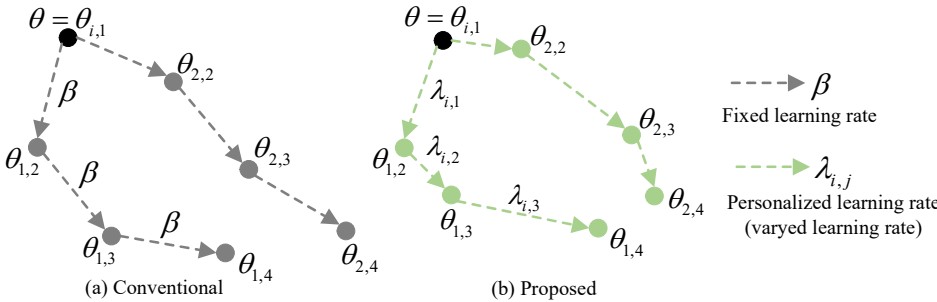

Figure 2: Overview of the inner-loop optimization using the personalized learning rates. (a) Conventional approaches employ the same learning rate for different tasks and across various stages of learning. (b) We take into account the dynamic changes in rapid adaptation within tasks and personalize the learning rate based on the current task's performance. This capability enables the model to better adapt to the complexity of tasks and the variability of data, ultimately leading to improved generalization.

After obtaining the personalized learning rates, the parameter updates for the inner loop are as follows:

$$\theta_{i,j+1} = \theta_{i,j} - \lambda_{i,j} \nabla_\theta \mathcal{L}_{\mathcal{T}_i}^{\mathcal{D}^s}(f_{\theta_{i,j}}) \tag{6}$$

The final parameter updates for the outer loop of the model are as follows:

$$\theta \leftarrow \theta - \alpha \nabla_\theta \frac{1}{B} \sum_{i=1}^{B} \mathcal{L}_{\mathcal{T}_i}^{\mathcal{D}^q}(\theta_{i,k} - \lambda_{i,k} \nabla_\theta \mathcal{L}_{\mathcal{T}_i}^{\mathcal{D}^s}(f_{\theta_{i,k}})) \tag{7}$$

---

**Algorithm 1** Meta-Learning with Personalized Learning Rates

---

**Require:** task distribution $p(\mathcal{T})$
**Require:** learning rates $\alpha$
1: Randomly initialize parameter $\theta$
2: **while** not done **do**
3:     Sample a batch of tasks $\{\mathcal{T}_1, \mathcal{T}_2, \cdots, \mathcal{T}_B\}$
4:     **for all** $\mathcal{T}_i$ **do**
5:         Initialize $\theta_{i,0} = \theta$
6:     **for** inner loop step j = 1 **to** k **do**
7:         Obtain the predicted values $y'$ and the ground truth values $y$
8:         Compute the information loss $\mathcal{L}_{KD_{i,j}} = \mathcal{D}_{KL}(p(y'_{i,j})\|p(y_i)))$
9:         Calculate the personalized learning rate:
$$\lambda_{i,j} = \begin{cases} lr_{max} - \mathcal{L}_{KD_{i,j}}, & if \mathcal{L}_{KD_{i,j}} \leq lr_{max} \\ lr_{min}, & if \mathcal{L}_{KD_{i,j}} > lr_{max} \end{cases}$$
10:         Inner loop parameter update $\theta_{i,j+1} = \theta_{i,j} - \lambda_{i,j} \nabla_\theta \mathcal{L}_{\mathcal{T}_i}^{\mathcal{D}^s}(f_{\theta_{i,j}})$
11:     **end for**
12:     Calculate query set loss $\mathcal{L}_{\mathcal{T}_i}^{\mathcal{D}^q}(f_{\theta_{i,k+1}})$
13:     **end for**
14:     Update meta model:
$$\theta \leftarrow \theta - \alpha \nabla_\theta \frac{1}{B} \sum_{i=1}^{B} \mathcal{L}_{\mathcal{T}_i}^{\mathcal{D}^q}(\theta_{i,k} - \lambda_{i,k} \nabla_\theta \mathcal{L}_{\mathcal{T}_i}^{\mathcal{D}^s}(f_{\theta_{i,k}}))$$
15: **end while**

---

## 4 EXPERIMENT

To thoroughly validate the efficacy of our approach, we conducted an extensive assessment across diverse scenarios. In the subsequent sections, we will furnish in-depth descriptions of the datasets

and experimental configurations employed in each scenario, along with the ultimate experimental outcomes. This exhaustive scrutiny will empower us to comprehensively gauge the performance of our method across a spectrum of tasks and datasets, affording us a deeper understanding of its adaptability and capacity for generalization. For additional experiments see Appendix A.1.

## 4.1 DATASETS

In our experiments, we utilized a diverse set of five datasets, namely CIFAR-FS (Bertinetto et al., 2018), miniImagenet (Vinyals et al., 2016), Aircraft (Yao et al., 2019), CUB (Caltech-UCSD Birds-200-2011) (Cimpoi et al., 2014), Dogs (Maji et al., 2013).

The CIFAR-FS dataset, derived from CIFAR-100 (Krizhevsky et al., 2009), is a classification dataset comprising 100 distinct classes representing various objects and concepts, including animals, objects, and people. Each class is composed of 600 color images, each measuring $32 \times 32$ pixels.

The miniImagenet dataset, a subset of ImageNet (Russakovsky et al., 2015), is tailored for few-shot classification tasks. It encompasses 100 different classes, similar to CIFAR-FS, each class consists of 600 color images, but with larger dimensions of $84 \times 84$ pixels.

The Aircraft is an aircraft classification dataset with 100 distinct aircraft classes, each containing approximately 100 images of varying sizes, angles, and perspectives. We standardized the image size to $84 \times 84$ pixels.

The CUB, dedicated to bird classification, encompasses 200 bird classes. Each class contains around 60 images of varying sizes, captured from different angles and environmental conditions. We resized all images to $84 \times 84$ pixels and selected 100 classes for our experiments.

The Dogs dataset is designed for dog classification, featuring 120 unique dog classes, each with about 100 images of varying sizes, poses, and backgrounds. We resized all images to $84 \times 84$ pixels.

## 4.2 IMPLEMENTATION DETAILS

In all our experiments, we employed a four-layer convolutional network as the backbone architecture. This network structure consists of four convolutional blocks, each comprising the following layers and parameter configurations: a convolutional layer (kernel size: 3x3, stride: 1, padding: 1, and 64 filters), a batch normalization layer, a ReLU activation layer, and a 2x2 maximum pooling layer. We selected the Adam optimizer with a learning rate of 0.001. Our task batch size was set to 4. During training, we utilized 5 gradient steps for parameter updates, while during evaluation, we performed 10 gradient steps for model evaluation. To ensure the reliability of our experimental findings, we report average accuracy with 95% confidence intervals, calculated across 600 randomly generated tasks from the test set. We maintained the reproducibility of our results by fixing the random seed at 1 and conducting all experiments under this seed.

## 4.3 FEW-SHOT CLASSIFICATION

In order to assess the adaptability of our approach, we conducted an examination of its performance when integrated with various meta-learning techniques. Specifically, we explored the impact of incorporating our method alongside optimization-based approaches such as ANIL, MAML, and FO-MAML. We quantitatively assessed the performance of these combinations on two well-established benchmark datasets, CIFAR-FS and miniImagenet, and summarized the outcomes in Table 1.

The consistency of experimental results underscores that our approach has achieved significant performance improvements when integrated with three distinct types of meta-learning algorithms. This enhancement can be attributed to our method's dynamic generation of negatively correlated learning rates during the training process, accomplished through the assessment of information loss between model predictions and ground truth. Our approach effectively adapts the learning rates in a task-specific manner based on the current task's performance.

During the training process, when information loss is low, it indicates that the model is performing well on the current task. To expedite parameter updates and the learning process, we increase the learning rate, facilitating faster model convergence and better adaptation to specific patterns and

Table 1: Average classification accuracy of few-shot classification on CIFAR-FS and miniImagenet. [†] denotes the local replication results.

| Model | CIFAR-FS | | miniImagenet | |
|---|---|---|---|---|
| | 5-way 1-shot | 5-way 5-shot | 5-way 1-shot | 5-way 5-shot |
| ANIL [†] | 55.67 ± 0.97% | 71.31 ± 0.76% | 47.77 ± 0.92% | 65.13 ± 0.75% |
| ANIL + MLPLR (ours) | **57.87 ± 0.96%** | **72.44 ± 0.75%** | **48.60 ± 0.92%** | **67.57 ± 0.74%** |
| MAML [†] | **56.44 ± 0.96%** | 72.74 ± 0.75% | 47.56 ± 0.92% | 65.54 ± 0.74% |
| MAML + MLPLR (ours) | 56.31 ± 0.96% | **74.57 ± 0.72%** | **50.31 ± 0.92%** | **67.97 ± 0.73%** |
| FOMAML [†] | **55.71 ± 0.96%** | 71.91 ± 0.74% | 45.90± 0.86% | 65.81 ± 0.75% |
| FOMAML + MLPLR (ours) | 55.50 ± 0.96% | **72.45 ± 0.73%** | **46.43 ± 0.90%** | **66.60 ± 0.72%** |

features within the task, thus enhancing training efficiency and performance. Conversely, when information loss is high, it signifies poorer model performance on the current task. To mitigate over-fitting and unstable parameter updates, we decrease the learning rate to reduce the rate of parameter changes, rendering parameter updates more stable and gradual while reducing the risk of overfitting.

These observations suggest that personalized learning rates represent a versatile technique capable of integration with various meta-learning algorithms to enhance their performance across diverse tasks. By adjusting learning rates on a task-specific basis during the training process, models become better equipped to adapt to new tasks and improve their classification performance. This approach provides an effective strategy for enhancing the robustness and adaptability of meta-learning algorithms.

### 4.4 FEW-SHOT FINE-GRAINED CLASSIFICATION

In contrast to few-shot classification, few-shot fine-grained classification necessitates the model to exhibit precise classification capabilities when confronted with tasks characterized by intricate class distinctions. Fine-grained classification tasks typically revolve around discriminating exceedingly subtle differences between categories, such as distinguishing between various dog or bird breeds. This imposes more significant demands on the model's learning capacity and its capacity for generalization in the context of few-shot fine-grained classification tasks.

To further substantiate the effectiveness of our model, we conducted experiments across three datasets featuring diverse fine-grained categories: Aircraft, CUB, and Dogs. The outcomes of these experiments are detailed in Table 2.

Table 2: Average accuracy of few-shot fine-grained classification on different datasets. [†] denotes the local replication results.

| Model | Aircraft | | CUB | | Dogs | |
|---|---|---|---|---|---|---|
| | 5-way 1-shot | 5-way 5-shot | 5-way 1-shot | 5-way 5-shot | 5-way 1-shot | 5-way 5-shot |
| ANIL [†] | 57.37±0.89 | **70.81±0.67** | 46.78±0.93 | 61.73±0.85 | 42.30±0.82 | **57.05±0.73** |
| ANIL + MLPLR (ours) | **59.97±0.91** | 70.67±0.66 | **51.31±0.95** | **64.18±0.82** | **44.08±0.83** | 56.95±0.73 |
| MAML [†] | 55.15±0.88 | 72.81±0.64 | 48.84±0.92 | 68.70±0.82 | 42.38±0.83 | 58.82±0.73 |
| MAML + MLPLR (ours) | **60.54±0.92** | **74.48±0.61** | **53.37±0.95** | **70.13±0.79** | **47.73±0.68** | **60.01±0.72** |
| FOMAML [†] | **54.05±0.90** | 68.42±0.69 | **49.96±0.95** | 60.33±0.83 | **41.56±0.81** | 56.20±0.73 |
| FOMAML + MLPLR (ours) | 53.90±0.89 | **71.52±0.64** | 46.18±0.93 | **65.08±0.85** | 40.25±0.81 | **59.73±0.74** |

Through a comprehensive analysis of the results in Table 2, we can observe a interesting phenomenon: regardless of the dataset used, the MAML + MLPLR method consistently demonstrates performance improvements across various scenarios. However, among different meta-learning methods, ANIL + MLPLR exhibits superior performance in 5-way 1-shot tasks, while FOMAML + MLPLR excels in 5-way 5-shot tasks.

This observation raises an intriguing question: why does ANIL + MLPLR seem to possess stronger learning and generalization capabilities in scenarios with limited samples, while FOMAML + MLPLR requires more samples to effectively engage in meta-learning? We can attribute these performance disparities to the distinct strengths and adaptabilities of different meta-learning methods when dealing with varying tasks and datasets.

This finding underscores the diversity of meta-learning approaches and reminds us to consider these differences when selecting methods that are suitable for specific tasks and datasets. In summary, the combination of personalized learning rates with appropriate meta-learning methods proves to be beneficial in optimizing the meta-learning process, ultimately leading to superior performance.

### 4.5 CROSS-DOMAIN FEW-SHOT CLASSIFICATION

In practical real-world applications, the need often arises to transfer a pre-trained model to new domains or tasks. To better gauge the practicality and versatility of our method, we undertake cross-domain few-shot classification assessments. In this context, we introduce the cross-domain few-shot classification scenario. Here, we engage in transfer learning within a 5-way 5-shot setting. During the training phase, we utilize the miniImagenet dataset to train our model. Subsequently, we apply the trained model to assess its performance on distinct domain datasets, including Aircraft, CUB, and Dogs. This evaluation serves as a means to evaluate the adaptability and generalization capabilities of our approach across a wide spectrum of domains and tasks.

Table 3: Average accuracy on 5-way 5-shot for cross-domain few-shot classification transferred from miniImagenet. $^{\dagger}$ denotes the local replication results.

| Model | miniImagenet $\rightarrow$ Aircraft | miniImagenet $\rightarrow$ CUB | miniImagenet $\rightarrow$ Dogs |
|---|---|---|---|
| ANIL $^{\dagger}$ | 35.25±0.54 | 50.94±0.66 | 44.87±0.64 |
| ANIL + MLPLR (ours) | **38.12±0.56** | **54.38±0.66** | **47.69±0.65** |
| MAML $^{\dagger}$ | 36.02±0.55 | 52.00±0.68 | 45.34±0.65 |
| MAML + MLPLR (ours) | **40.38±0.58** | **54.38±0.70** | **49.15±0.70** |
| FOMAML $^{\dagger}$ | 37.11±0.57 | 52.94±0.70 | 43.88±0.62 |
| FOMAML + MLPLR (ours) | **39.01±0.60** | **54.47±0.71** | **49.72±0.67** |

According to the findings extracted from Table 3, our methodology has consistently demonstrated noteworthy enhancements in performance. A discerning observation reveals that, in comparison to the other two datasets, the transfer performance on the Aircraft dataset appears to exhibit a relatively suboptimal performance trend. This discrepancy can be principally ascribed to the composition of the miniImagenet dataset, which encompasses categories associated with birds and dogs but notably lacks categories pertaining to aircraft. Consequently, as the model is transposed from the familiarity of miniImagenet to the unfamiliar terrain of the Aircraft dataset, a conspicuous domain disparity emerges, which, may contribute to the observed performance decrement.

## 5 CONCLUSION

In this paper, we propose a novel meta-learning approach called MLPLR. This method overcomes the limitation of fixed learning rates in traditional meta-learning methods. In MLPLR, we dynamically adjust the learning rates using the information loss between task predictions and ground truth, enabling them to adapt autonomously based on the performance of the current task. Encouragingly, the MLPLR method not only collaborates effectively with various meta-learning models but also requires relatively fewer additional computational resources. We conducted extensive experiments on five widely used meta-learning datasets to demonstrate the compatibility and effectiveness of our approach. These experimental results not only validate the applicability of MLPLR but also underscore its potential in enhancing meta-learning performance. The introduction of MLPLR presents a flexible and efficient new approach to the field of meta-learning, offering robust support for future research and applications.

### REPRODUCIBILITY STATEMENT

To ensure the thorough validation and reproducibility of our research outcomes, we have included comprehensive code within the supplementary materials. This code encompasses the entire experimental workflow, including aspects such as model construction, model training and evaluation, hyperparameter configuration, and more.

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

# A APPENDIX

## A.1 FEW-SHOT CLASSIFICATION

We deliberated upon the influence of varying class quantities on model accuracy. As depicted in Figure 3, the results demonstrate that as the number of classes increases, the efficacy of personalized learning rates becomes more pronounced. With the escalation in class count, discernible disparities may manifest among different tasks, potentially leading to divergent convergence rates during the training process. In such circumstances, the adoption of personalized learning rates facilitates the model in better accommodating the idiosyncrasies of each task, allowing for tailored adjustments according to the requisites of each task, thus enhancing the overall performance of the model.

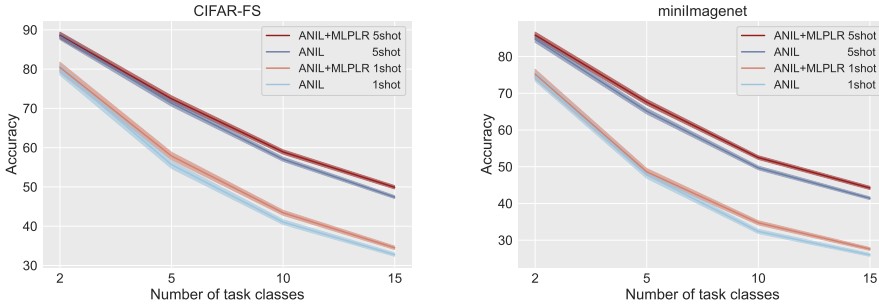

Figure 3: Classification results for different numbers of task classes.

