# OpenReview forum: "Meta-Learning with Personalized Learning Rates for Rapid Task Mastery"
_ICLR.cc/2024/Conference — Submitted to ICLR 2024_

### Official Review · Reviewer_RAUk · 2023-10-30

**Soundness:** 2 fair
**Presentation:** 2 fair
**Contribution:** 2 fair
**Rating:** 3
**Confidence:** 4

**Summary:**

The paper proposes a new method for generating adaptive learning rates without requiring additional computational resources. This method, referred to as Meta-Learning with Personalized Learning Rates adjusts based on information loss evaluation between the model's predictions and the ground truth. Advantages of this approach include its independence from extra computational resources, thus making it applicable in various computing environments without increased costs, and its ability to be integrated seamlessly with various meta-learning models. The method enhances model performance across diverse tasks and data conditions, making the training process more efficient and intelligent, and reducing the need for hyperparameter tuning. Extensive testing indicates the wide applicability of the method, enhancing performance irrespective of model constraints.

**Strengths:**

+ This paper an innovative approach to the dynamic learning rate scheme along the within-task adaptation steps, while the existing works primarily considered either task-wise learning rates (Yu et al., 2021; Kim et al., 2022) or global learning rates (Li et al., 2017; Behl et al., 2019; Zou et al., 2021).

+ The proposed method has demonstrated its effectiveness in the cross-domain few-shot classification tasks as well as the number of classes per task increases.

**Weaknesses:**

-  This paper proposes a strategy that derives learning rates from the Kullback-Leibler (KL) loss/ information gain. However, this novel, empirical-focused method could introduce potential issues. One possible concern is that loss and learning rate are different things, raising
 the need for a normalized function to map the loss to a real number. Another issue is that the loss can vary dramatically. To constrain this, the authors have brought two extra hyperparameters $lr_{\max}$ and $lr_{\min}$ as the constraints. This strategy is somehow primitive and naive. For one possible technical issue please refer to question 1.

- No normalization is present in the Eqn (4). It will bring another problem that the learning rate will change as batch size changes. The adaptive learning rate also does not follow the Linear Scaling Rule.

- The justification for the principle of the proposed method appears to be insufficient. In the paper, the authors explain the motivation as the task performance. If in that case, why would we choose other measures, such as CrossEntropy? It would be better if the author could provide some theoretical insights in a convex setting (Behl et al., 2019; Zou et al., 2021). Alternatively, the author may need to strengthen the motivation with more explainable observations.

- The novelty is limited. The main contribution of this paper is to propose a meta-learning framework with adaptive learning rates while (Li et al., 2017) have considered it a learnable module, diminishing the originality of the proposed approach. The motivation is not convincing enough "Information loss is high, indicating poor model performance on the current task, we reduce the learning rate to slow down parameter changes to avoid overfitting and unstable parameter updates". I think it would be better to build it from the information-gain standpoint. The proposed method simply uses the KL loss as the learning rate with some technical flaws that limit the contribution. The absence of an explanation or justification of the underlying principles further weakens the method's impact.

**Questions:**

1. In Eqn (5), if $\mathcal{L}\_{KD} > lr_{\max}$, why we choose a larger learning rate $lr_{\min} > 0$ than if $\mathcal{L}\_{KD}=lr_{\max}$?

2. How we determine hyperparameter $lr_{\max}, lr_{\min}$? If we select a small $lr_{\max}$, the proposed method will reduce to vanilla MAML. If we select a big $lr_{\max}$, then the update step size may be large enough.

3. Why don't compare with other measures?

---

### Official Review · Reviewer_8ubD · 2023-11-01

**Soundness:** 2 fair
**Presentation:** 2 fair
**Contribution:** 1 poor
**Rating:** 3
**Confidence:** 4

**Summary:**

The author proposed an adaptive learning rate adjustment approach to improve convergence and adaptation in meta learning.

**Strengths:**

The authors conducted pretty detailed evaluation on the proposed approach with detailed analysis on the result.

**Weaknesses:**

Novelty is limited. The idea is not new. Numerous works have proposed approaches similar to the "personalized learning rate" proposed in this work, in the name of "learnable learning rate", "hyper-gradient" etc. Some examples including Meta-SGD [1] which learns per parameter learning rates to accelerate the training process in meta learning, and [2] which learn the per parameter gradient update rule for meta-learning.

For works on improving convergence of machine learning algorithms, it is better to have theoretical analysis on the convergence speed, e.g.  evaluation on the shrinkage of distance to optimal parameters. The current work is relatively weak on this aspect.

The information loss that used to decide learning rate is just KL-divergence, would be better if the author could try more options on the information loss.


[1] Meta-sgd: Learning to learn quickly for few shot learning, 2017
[2] Gradient-based meta-learning with learned layerwise metric and subspace, 2018

**Questions:**

The information loss that used to decide learning rate is just KL-divergence, would be better if the author could try more options on the information loss.

---

### Official Review · Reviewer_wr36 · 2023-11-08

**Soundness:** 1 poor
**Presentation:** 1 poor
**Contribution:** 1 poor
**Rating:** 3
**Confidence:** 5

**Summary:**

The paper introduces MLPLR, a new approach that dynamically adjusts learning rates based on task performance, overcoming fixed rate limitations in traditional methods. MLPLR dynamically adjusts learning rates based on the information loss between task predictions and ground truth, allowing for autonomous adaptation according to task performance.
It collaborates effectively with various meta-learning models, requires fewer computational resources, and demonstrates potential for enhancing meta-learning performance across multiple datasets.

**Strengths:**

(1) The idea of dynamically generating customized learning rates seems promising.

(2) Achieving good performance across various optimization-based meta-learning tasks with the proposed MLPLR is notable.

**Weaknesses:**

(1)  The paper suffers from issues related to clarity and precision in writing. There is a need for improved writing quality and rigor. For instance, in the abstract, the statement about "traditional meta-learning" is overly broad, as meta-learning encompasses various methods, and this paper specifically addresses optimization-based meta-learning. Furthermore, the introduction section provides an excessive amount of background information, which could be condensed to allow for a more substantial presentation of the proposed method. Additionally, although Meta-SGD is mentioned as a related method that dynamically generates learning rates, there is limited discussion and comparison with it in the paper.

(2)  The use of information loss for dynamically adjusting learning rates is not novel, as it has been employed in several prior meta-learning methods [1]. Consequently, the paper lacks significant technical innovation. Moreover, there is a lack of thorough discussion regarding the choice of using KL divergence loss to generate learning rates instead of directly utilizing cross-entropy loss, which should be elucidated for a more comprehensive understanding.

[1] For example, the use of information loss can be found in "IEPT: Instance-Level and Episode-Level Pretext Tasks for Few-Shot Learning" (ICLR21).

(3) The paper lacks essential experimental details. For instance, the definition and treatment of hyperparameters such as lr_max and lr_min in Equation (5) need clarification. It is unclear whether these values are hyperparameters or if they are determined through some adaptive mechanism. The absence of experiments examining the performance across different hyperparameter settings diminishes the comprehensiveness of the paper. Additionally, the paper lacks a strong theoretical foundation and motivation for the specific design choices regarding the learning rate adaptation, which should be addressed for a more robust contribution.

**Questions:**

Many of the concerns raised in the weaknesses section overlap with the following questions:

What is the appropriate way to determine lr_max and lr_min in Equation (5)?

The paper's writing quality needs enhancement, and it could benefit from revisions.

I believe that the current version may face challenges in meeting the acceptance criteria for ICLR.

---

### Meta-Review · Area_Chair_u5Jk · 2023-12-16

**Metareview:**

The paper proposes a meta-learning approach with adaptive learning rates based on task performance, overcoming fixed rate limitations in traditional methods.

All the reviewers and AC agree that the paper lacks (1) technical novelty, as the proposed approach to dynamically adjust learning rate has been studied in prior meta-learning approaches under the umbrella of adaptive/learnable learning rates – see all Reviewers comments,  (2) clarity and rigor in presentation -- see Reviewer wr36 concerns and suggestions on how to improve, (3) an explanation or justification of the underlying principles, e.g. KL-divergence loss versus cross-entropy loss– see all Reviewers comments.
There is no author response for this paper. A general consensus among reviewers and AC suggests, in its current state the manuscript is not ready for a publication. We hope the reviews are useful for improving and revising the paper.

**Justification For Why Not Higher Score:**

All reviewers and AC are in consensus about rejection.

**Justification For Why Not Lower Score:**

N/A

---

### Decision · Program_Chairs · 2024-01-16

Reject